# Decreased risk of Parkinson's disease in diabetic patients with thiazolidinediones therapy: An exploratory meta-analysis

**Yueli Zhu, Jiali Pu, Yanxing Chen\*, Baorong Zhang**\*

Department of Neurology, the Second Affiliated Hospital, School of Medicine, Zhejiang University, Hangzhou, China

\* brzhang@zju.edu.cn (BZ); chenyanxing@zju.edu.cn (YC)

## Abstract

### Background

It has been found that thiazolidinediones (TZDs) may play a protective role in animal models of Parkinson's disease (PD), while the results remain controversial whether TZDs protect against Parkinson's disease in humans. The purpose of this meta-analysis is to explore the association between TZDs use and the incidence of PD in diabetic patients.

### Methods

A systematic online search was conducted to find studies published up to 31 December 2018. In our exploratory meta-analysis, studies comparing incidence of PD between TZD-treated and non-TZD-treated groups of diabetic patients were included. Data analysis was performed using a random or fixed effects model and expressed as odds ratios (OR) with 95% confidence interval (95% CI). We used the Cochrane Collaboration's Review Manager 5.3 software to analyze data.

### Results

In total, 5 retrospective observational cohort studies were identified which met the inclusion criteria. The pooled odds ratio (OR) was 0.70 [95% CI, 0.51 to 0.96; $p = 0.03$] in a random-effects model, indicating a 30% lower risk of developing PD in diabetic patients treated with TZDs compared with non-TZD-treated patients.

### Conclusion

In this exploratory meta-analysis, we found that TZDs use was associated with reduced risk of PD in diabetic patients. However, this meta-analysis was not registered online although we followed a protocol designed for it. Further prospective observational studies with larger sample size and more strict inclusion criteria including controlling for diabetes complication severity index, hypoglycemic drugs combination, sex ratio, and comorbidity are needed to guide whether RCTs are warranted. And RCTs can better determine whether TZDs use could lower incidence of PD in diabetic patients.

**Data Availability Statement:** All relevant data are within the paper and its Supporting Information files.

**Funding:** The design of the study was supported by the National Natural Science Foundation of China (81520108010). The data collection and analysis were supported by the National Natural Science Foundation of China (81870826). The interpretation was supported by the National Natural Science Foundation of China (81771216). The manuscript writing was supported by Zhejiang Provincial Natural Science Foundation of China (LY18H090004).

**Competing interests:** The authors have declared that no competing interests exist.

## Introduction

Parkinson's disease (PD) is a prevalent and complex neurodegenerative disorder with prominent loss of dopaminergic neurons in the substantia nigra. Rest tremor, rigidity, bradykinesia, and postural instability are the four classical motor symptoms [1]. PD is the second most common neurodegenerative disease. The annual incidence is 14/100 000 of the total population in Europe and the Americas, while the rate increased to 160/100 000 in people aged over 65 years. The incidence is similar in Asia [2].

Thiazolidinediones (TZDs), a class of peroxisome proliferator-activated receptor-gamma (PPAR-γ) agonists, can improve insulin sensitivity and lower blood glucose level for patients with type 2 diabetes [3]. Only pioglitazone and rosiglitazone are currently in clinical use since troglitazone was withdrawn due to hepatotoxicity [3]. Recently, accumulating experiments indicated that TZDs could exert anti-inflammatory and neuroprotective effects in a series of PD animal models [4–7]. Nevertheless, one randomized controlled trial (RCT) indicated that pioglitazone is unlikely to be able to slow the disease progression in early PD patients without diabetes [8]. To date, several retrospective observational cohort studies have assessed the association between the TZDs use and the incidence of PD in diabetic patients, but with controversial results [9–13]. Although RCT studies are considered more reliable to confirm the treatment effectiveness, observational studies can overcome the limited generalizability and inadequate follow-up to identify rare adverse effects of RCTs, the outcomes of which can help guide the future RCT design.

Hence, this exploratory meta-analysis was conducted to evaluate the efficacy of TZDs in reducing PD risk among diabetic patients, though very limited number of high quality studies are available due to the heterogeneity of disease phenotypes and treatment approaches both in diabetes mellitus and PD. To the best of our knowledge, this is the first meta-analysis assessing the association between TZDs use and PD incidence.

## Materials and methods

### Search strategy

A comprehensive literature search was conducted up to 31 December 2018. The Cochrane Central Register of Controlled Trials (CENTRAL), PubMed, Web of Science and Embase were screened for eligible studies by two independent investigators (Yueli Zhu, Yanxing Chen) and a coefficient of agreement (Kappa) was calculated. Disagreements were solved through discussion with a third reviewer if necessary (Jiali Pu).

All methods were specified in advance in a study protocol, which is available in S1 File. And we also deposited our protocol in protocols.io (the DOI link: http://dx.doi.org/10.17504/protocols.io.7k8hkzw). This meta-analysis was conducted according to the Preferred Reporting Items for Systematic Reviews and Meta-Analysis statement (PRISMA) guidelines (S1 Table) [14]. We identified all potential studies using combinations of the following terms (free text and/or medical subject headings (MeSH) terms adapted to the requirements of each database): (Parkinson's disease OR Parkinson disease OR PD OR parkinsonism OR paralysis agitans) and (thiazolidinediones OR glitazones OR pioglitazone OR rosiglitazone). The full search strategy is detailed in S2 File. We also manually searched the references of reviews and included articles to identify additional relevant studies.

### Inclusion and exclusion criteria

The studies eligible for this meta-analysis met all of the following criteria: (1) studies that investigated both TZDs use and incidence of PD among diabetic patients; (2) comparison of TZDs with non-TZD treatments; (3) the diagnosis of PD as an outcome.

Studies were excluded for the following reasons: (1) case reports, comments, reviews, editorials and clinical guidelines; (2) studies that did not provide valid data, such as the diagnosis of PD was not the outcome or the data was incomplete. We also contacted the corresponding authors by email to obtain primary outcome data.

### Data extraction

Two investigators (Yueli Zhu, Yanxing Chen) independently extracted the relevant data from each study using a detailed form. We extracted the data from each study including first author's name, publication year, location, TZDs type, sample size, follow-up duration, definition of PD, the incidence of PD. If necessary, disagreements were solved through discussion with a third reviewer (Jiali Pu).

### Quality assessment of cohort studies

Two investigators (Yueli Zhu, Yanxing Chen) independently evaluated all of the retrieved studies. The risk of bias assessment of included studies were evaluated according to the Risk Of Bias In Non-randomized Studies-of Interventions (ROBINS-I tool). This tool is based on seven domains that included bias due to confounding, selection of participants, classification of interventions, deviations from intended interventions, missing data, measurement of outcomes and selection of the reported result [15]. "Low risk", "Moderate risk", "Serious risk" and "Critical risk" are categorized for risk of bias judgments [15]. The study with low risk of bias for all domains was judged to be at a low rating; the study with low or moderate risk of bias for all domains was judged to be at a moderate rating; the study with a serious risk of bias in at least one domain was judged to be at a serious rating; and the study with a critical risk of bias in at least one domain was judged to be a critical rating [15]. Disagreements were solved through discussion with a third reviewer (Jiali Pu).

### Statistical analysis

We performed all statistical analysis using the Review Manager 5.3 software (The Nordic Cochrane Centre, The Cochrane Collaboration, Copenhagen) and set odds ratio (OR) at a 95% confidence interval (95% CI) to assess the association between TZDs use and the incidence of PD in diabetic patients.

We estimated the heterogeneity between studies by performing both the chi-square test (assessing the $P$-value) and the $I^2$ statistic. $P < 0.1$ and $I^2 > 50\%$ for heterogeneity were considered with significant differences, then a random-effects model was used to calculate the pooled ORs. And outcome data were pooled using a fixed-effects model with insignificant heterogeneity. The between-study variance was estimated using tau-squared ($t^2$ or $Tau^2$) statistic in the random-effects model. In order to find out the potential sources of heterogeneity and assess the robustness of results, sensitivity analysis was performed by omitting one study per iteration. In addition, we conducted subgroup analysis by using TZDs type, ethnicity, and follow-up duration as stratifying variables. Funnel plots were conducted to assess the latent publication bias. When the shape of funnel plots was asymmetric, possible publication bias was determined.

## Results

### Search results and study characteristics

A total of 113 potential relevant records were indentified initially according to our literature search strategy, of which 63 were excluded due to duplication. After carefully screening the

titles and abstracts, we excluded 44 studies which did not meet eligibility criteria. One study was excluded because of failure to be related to the incidence of PD. Ultimately, the remaining five qualified studies [9–13] involving 347,556 patients were considered suitable for this meta-analysis. The Kappa for agreement between the two independent investigators was 0.8. The screening process is displayed in a flow diagram (Fig 1), and the main characteristics of these studies are summarized in Table 1. The 5 studies were all cohort studies, of which, 3 targeted at White patients and the other 2 were from Asia. Four studies compared TZD-treated with non-TZD-treated patients, whereas the other one compared pioglitazone-treated with non-pioglitazone-treated patients. The follow-up time lasted from 5 to 15 years. The ROBINS-I tool was applied for quality assessment (Table 2). All 5 studies were assessed to be at an overall rating of moderate risk of bias.

## Exploratory meta-analysis of incidence of PD

Three [9,10,12] studies showed that TZDs use was associated with a reduced incidence of PD, whereas the other 2 [11,13] studies did not find such association. Pooled analysis of the 5

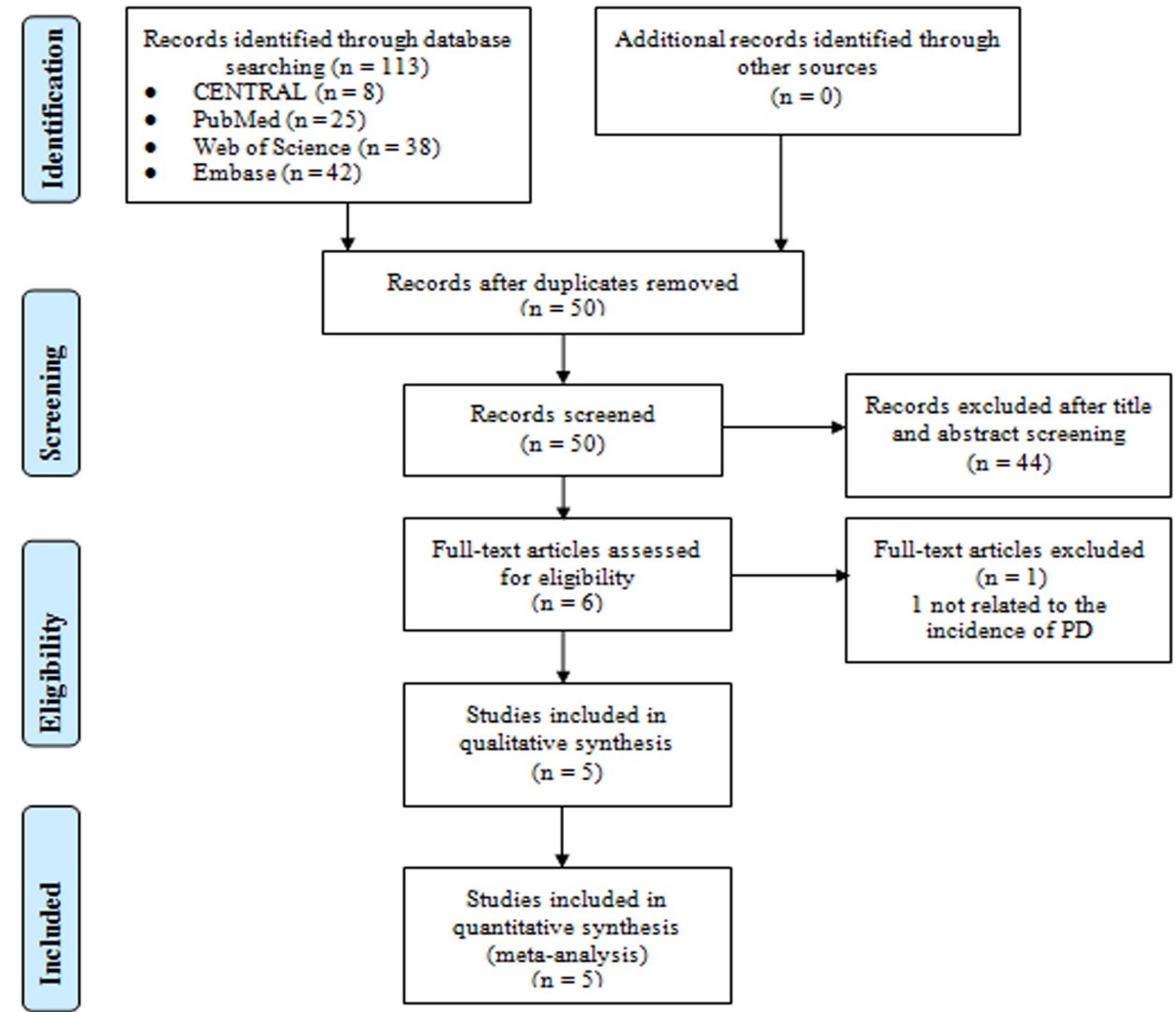

**Fig 1. Flow chart of the literature search and selection process.**

**Table 1. The characteristics of 5 included cohort studies.**

| Study | Region (ethnicity) | Medication studied | | Total participants (events) | | Follow-up time | PD ascertainment |
|---|---|---|---|---|---|---|---|
| | | Exposed group | Comparison group | Exposed group | Comparison group | | |
| Brakedal et al., 2017 [9] | Norway (White) | TZD | Metformin | 8396 (57) | 94349 (938) | 10 years | ICD-10, ICPC-2 |
| Brauer et al., 2015 [10] | UK (White) | TZD | No TZD Use | 44597 (175) | 120373 (517) | 15 years | Clinical records |
| Connolly et al., 2015 [11] | USA (White) | TZD | sulfonylurea | 5225 (27) | 20283 (129) | 9 years | ICD-9 |
| Lin et al., 2018 [12] | Taiwan (Asian) | TZD | No TZD Use | 8250 (52) | 30271 (492) | 12 years | ICD-9 |
| Wu et al., 2018 [13] | Taiwan (Asian) | Piogli-tazone | No piogli-tazone use | 7906 (119) | 7906 (138) | 5 years | ICD-9 |

ICD, international classification of diseases; ICPC, International Classification of Primary Care.

included studies showed that the OR was statistically significant (OR, 0.70; 95% CI, 0.51 to 0.96; $p = 0.03$; Fig 2A), in spite of obvious heterogeneity (Chi$^2$, 28.25; $I^2 = 86\%$; $p < 0.0001$; tau$^2 = 0.10$), indicating that the use of TZDs was associated with significantly reduced risk of PD in diabetic patients with TZDs treatment compared with those without.

## Sensitivity analysis

Sensitivity analysis was conducted to better explore the potential heterogeneity by removing one study at a time and checking the consequent effects. When we deleted the study conducted by Lin et al. [12], the heterogeneity decreased dramatically to 10%. When we used the fixed-effects model, a significant association of TZDs use with lower incidence of PD in diabetic patients was observed (OR, 0.84; 95% CI, 0.74 to 0.94; $p = 0.003$; $I^2 = 10\%$; Fig 2B), consistent with the analysis when all 5 studies combined.

## Subgroup analysis

We conducted a subgroup analyses according to TZDs type, ethnicity and follow-up duration (Table 3). With regard to different TZDs, pioglitazone use [13] (OR, 0.86; 95% CI, 0.67 to 1.10;

**Table 2. The quality of the included studies assessed by ROBINS-I.**

| Study | Bias due to cofounding | Bias in selection of participants into the study | Bias in classification of interventions | Bias due to deviations from intended interventions | Bias due to missing data | Bias in measurement of outcomes | Bias in selection of the reported result | Overall Assessment |
|---|---|---|---|---|---|---|---|---|
| Brakedal et al., 2017 [9] | Moderate | Low | Low | Low | Low | Low t | Low t | Moderate |
| Brauer et al., 2015 [10] | Moderate | Low | Low | Low | Low | Low | Moderate | Moderate |
| Connolly et al., 2015 [11] | Moderate | Low | Low | Low | Low | Low | Low t | Moderate |
| Lin et al., 2018 [12] | Moderate | Low | Low | Low | Low | Low | Low t | Moderate |
| Wu et al., 2018 [13] | Moderate | Low | Low | Low | Low | Low | Low t | Moderate |

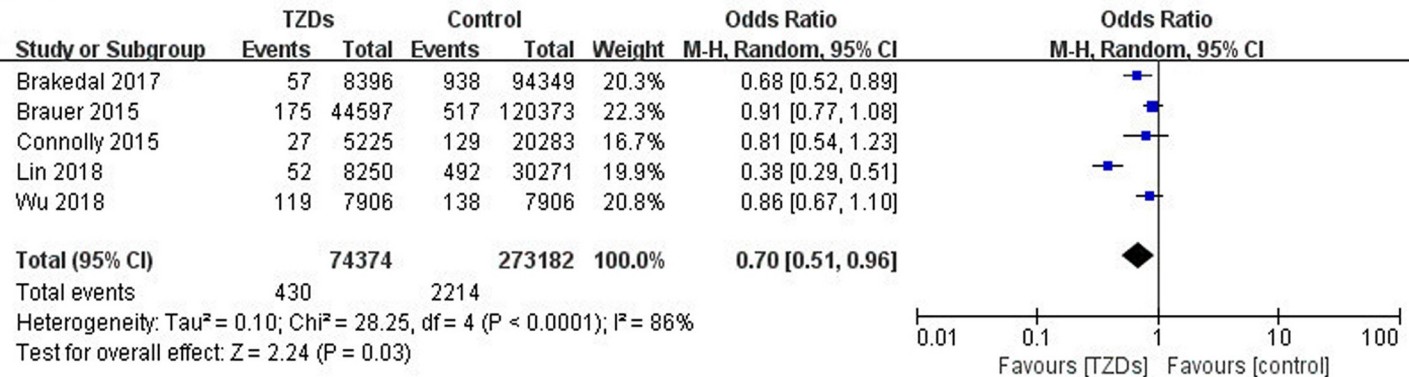

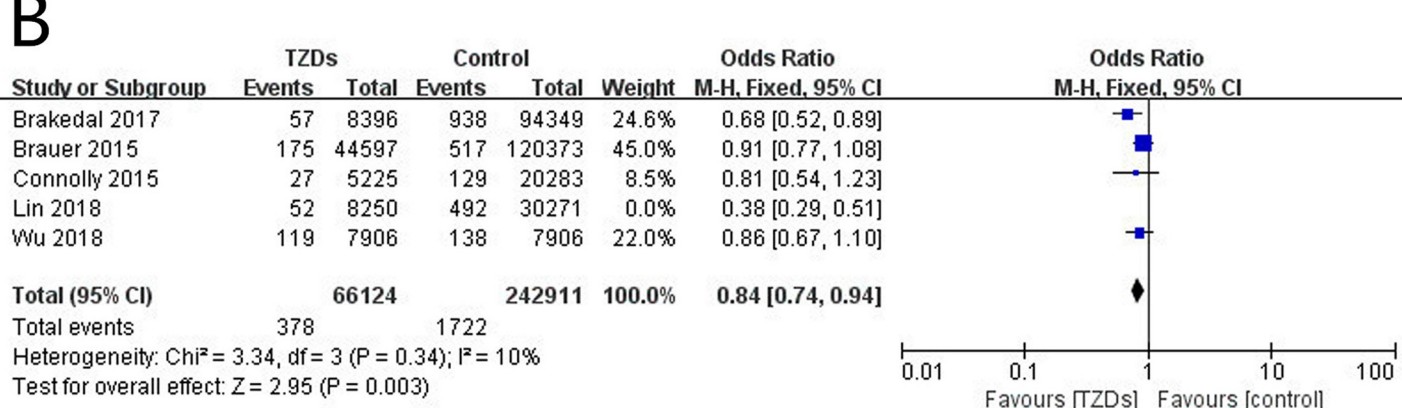

**Fig 2.** Forest plots for (A) Incidence of PD in TZD-treated versus non-TZD-treated groups in diabetic patients of all five cohort studies; (B) Incidence of PD in TZD-treated versus non-TZD-treated groups in diabetic patients after removing the study by Lin et al. Each study is represented by a block at the point estimate of intervention effect. There is no association between TZDs use and PD incidence when the block is the on the vertical line; TZDs use is associated with reduced incidence of PD when the block is on the left of the vertical line; TZDs use is associated with increased incidence of PD when the block is on the right of the vertical line. Events, the number of patients diagnosed with PD at follow-up; Total, the total number of patients in each study; M-H, Mantel-Haenszel. CI, confidence interval.

**Table 3. Subgroup analyses for the effect of TZDs use on incidence of PD in diabetic patients.**

| Subgroups | Number of studies | OR (95%CI) | Heterogeneity | | Significance test | |
|---|---|---|---|---|---|---|
| | | | $I^2$ (%) | $p$ | z | $p$ |
| TZDs | | | | | | |
| Only pioglitazone | 1 | 0.86 (0.67, 1.10) | NA | NA | 1.19 | 0.23 |
| Any kind of TZDs | 4 | 0.66 (0.44, 0.99) | 89 | <0.00001 | 2.00 | 0.05 |
| Ethnicity | | | | | | |
| White | 3 | 0.83 (0.72, 0.95) | 39 | 0.19 | 2.70 | 0.007 |
| Asian | 2 | 0.58 (0.26, 1.28) | 94 | <0.0001 | 1.35 | 0.18 |
| Follow-up duration | | | | | | |
| < 10 years | 2 | 0.85 (0.68, 1.05) | 0 | 0.81 | 1.54 | 0.12 |
| ≥ 10 years | 3 | 0.63 (0.38, 1.03) | 92 | <0.00001 | 1.83 | 0.07 |

NA, not applicable; TZDs, thiazolidinediones; OR, odds ratio; CI confidence interval.

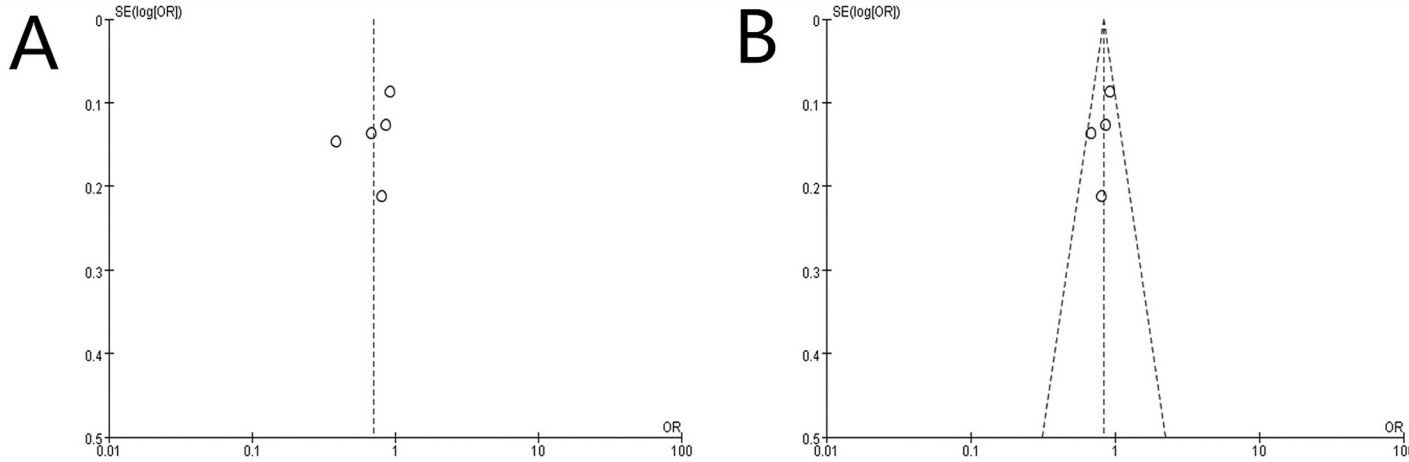

**Fig 3.** Funnel plots for publication bias for (a) Incidence of PD of all five cohort studies; (b) Incidence of PD of four cohort studies after removing the study by Lin et al. It is worth noting that power of the tests is low, which might not be able to distinguish chance from real asymmetry, since the number of included studies is less than ten.

$p = 0.23$) was not associated with the incidence of PD, whereas TZDs use [9–12] was associated with a decreased PD incidence (OR, 0.66; 95% CI, 0.44 to 0.99; $p = 0.05$; $I^2 = 89\%$). Only one study assessed the use of pioglitazone, therefore, the results should be interpreted with caution. However, the heterogeneity was still high, indicating that TZDs type was not the source of heterogeneity. Regarding different ethnicity, significant differences were found in 3 studies [9–11] among White patients (OR, 0.83; 95% CI, 0.72 to 0.95; $p = 0.007$; $I^2 = 39\%$) while the association between TZDs use and PD incidence was not found in the other 2 studies [12,13] among Asian patients (OR, 0.58; 95% CI, 0.26 to 1.28, $p = 0.18$; $I^2 = 94\%$). The heterogeneity in the White ethnic group decreased obviously ($I^2 = 39\%$) while $I^2$ in Asian groups increased to 94%. Hence, ethnicity might be one of the sources of heterogeneity. When follow-up duration was enrolled as a grouping feature, there were no significant differences in PD incidence between TZDs use and non-TZD treatments in the 2 studies [11, 13] with a follow-up duration $< 10$ years (OR, 0.85; 95% CI, 0.68 to 1.05, $p = 0.12$; $I^2 = 0\%$) and the other 3 studies [9,10,12] with a follow-up duration $\geq 10$ years (OR, 0.63; 95% CI, 0.38 to 1.03; $p = 0.07$; $I^2 = 92\%$). $I^2$ for studies with follow-up duration $< 10$ years was 0, and it increased to 92% in studies with follow-up duration $\geq 10$ years, indicating that follow-up duration might be one of the sources of heterogeneity.

## Publication bias

Publication bias was tested with funnel plots (Fig 3). The shape of funnel plots revealed no obvious asymmetry, indicating that there was no significant publication bias among these studies.

## Discussion

TZDs are selective and potent agonists for PPAR-γ, which can improve insulin resistance and glycemia for patients with type 2 diabetes [3]. Studies have shown that TZDs have neuroprotective properties in preclinical models of PD [4–7]. But, it remains controversial whether TZDs could benefit diabetic patients, leading to reduced risk of PD.

To the best of our knowledge, this is the first meta-analysis demonstrating the relationship between TZDs use and the incidence of PD in diabetic patients. Our exploratory meta-analysis

showed that the use of TZDs was associated with significantly reduced risk of PD. To find out the sources of heterogeneity and assess the robustness of the results, the sensitivity analysis was then performed. After removing the study conducted by Lin et al. [12], the heterogeneity decreased dramatically to 10% and the result remained unchanged. We speculate that heterogeneity might be derived from this study. Based on analysis, we found that Lin et al. [12] reported the lowest OR among 5 studies. PD is a common age-related neurodegenerative disease and its prevalence increases steadily with age [16]. A lower incidence of PD was observed in females than in males due to higher oestrogen activity, which leads to higher striatal dopamine levels [17]. In the study conducted by Lin et al., we found more female and younger patients in TZD group than those in the non-TZD group. Hence, age and sex differences may lead the study by Lin et al. [12] to be the source of heterogeneity. Besides, TZDs type, ethnicity, and follow-up duration were used as stratifying variables in subgroup analysis. While classified by ethnicity and follow-up duration, the degree of subgroup heterogeneity reduced obviously, indicating that ethnicity and follow-up duration might be the sources of heterogeneity.

Previous studies investigating the potential association between TZDs treatment and PD risk turned out to be contradictory. We performed an exploratory meta-analysis to overcome the limitation of limited sample size and further analyzed the potential role of TZDs use in influencing PD susceptibility. Our results imply a decreased risk of PD with TZDs use. The precise mechanism remains uncertain. PPAR-γ has been shown to attenuate neuronal damage by reducing neuroinflammation in animal models of PD [4–7]. According to the study by Simuni et al. [8], TZDs is unlikely to modify disease progression in early PD. However, our meta-analysis provides supportive data for the use of TZDs as a preventive strategy for PD in diabetic patients and its potential role as a therapeutic drug for PD.

There are some limitations that should be considered when interpreting the results of the exploratory meta-analysis. Firstly, we followed a protocol designed for the meta-analysis, however, it was not registered online. Secondly, according to the criteria outlined in the Cochrane handbook, the power is low which might make the funnel plots not be able to distinguish chance from real asymmetry when the included studies are less than ten in this meta-analysis. Thirdly, only five related studies were identified. Future prospective studies with larger cohorts and more strict inclusion criteria with respect to controlling for diabetes complication severity index, hypoglycemic drugs combination, sex ratio, and comorbidity are of great value to confirm these results. And by taking these confounding variables into consideration, the outcome could provide more information for the designation of RCTs. Further biological studies are also needed to specify the potential neuroprotective mechanism of TZDs in PD. Fourthly, patients in the prodromal stage of PD are difficult to be identified and may be enrolled, which can not be avoided. And it can lead to the different medication use and overall treatment differences between diabetics in the prodromal stage of PD and diabetics without signs of PD. Therefore, it is important to take into account the non-motor symptoms which may indicate the prodromal PD. Fifthly, different TZDs doses should be explored to find the optimal protective effect, which was unable to achieve in our exploratory meta-analysis due to the absence of data.

## Conclusion

This exploratory meta-analysis supports the value of using TZDs in reducing PD incidence in diabetic patients. A definite conclusion needs further confirmation. Future prospective observational cohort studies with larger cohorts and more strict inclusion criteria including controlling for diabetes complication severity index, hypoglycemic drugs combination, sex ratio, and comorbidity are of great value to further confirm the association between TZDs use and PD

incidence. Meanwhile, the information including detailed TZDs type and doses should be publicly available in future studies to better identify the optimal TZDs doses of reducing PD incidence in diabetic patients.

## Supporting information

**S1 File. Study protocol.**
(DOC)

**S2 File. Search strategy.**
(DOC)

**S1 Table. PRISMA checklist.**
(DOC)

## Author Contributions

**Conceptualization:** Yueli Zhu, Jiali Pu.

**Investigation:** Yueli Zhu, Yanxing Chen.

**Methodology:** Yueli Zhu, Yanxing Chen.

**Software:** Yueli Zhu, Jiali Pu, Yanxing Chen.

**Writing – original draft:** Yueli Zhu.

**Writing – review & editing:** Yanxing Chen, Baorong Zhang.

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
