## [Decision Letter · Decision Letter 0]

12 Aug 2019

PONE-D-19-19116

Decreased risk of Parkinson’s disease in diabetic patients with thiazolidinediones therapy: A meta-analysis

PLOS ONE

Dear Dr. Baorong Zhang,

Thank you for submitting your manuscript to PLOS ONE. After careful consideration, we feel that it has merit but does not fully meet PLOS ONE’s publication criteria as it currently stands. Therefore, we invite you to submit a revised version of the manuscript that addresses the points raised during the review process.

ACADEMIC EDITOR: The reviewers have raised a number of points which we believe major modifications are necessary to improve the manuscript, taking into account the reviewers' remarks.  Please consider and address each of the comments raised by the reviewers before resubmitting the manuscript. This letter should not be construed as implying acceptance, as a revised version will be subject to re-review.

We would appreciate receiving your revised manuscript by Sep 26 2019 11:59PM. To enhance the reproducibility of your results, we recommend that if applicable you deposit your laboratory protocols in protocols.io, where a protocol can be assigned its own identifier (DOI) such that it can be cited independently in the future. For instructions see: http://journals.plos.org/plosone/s/submission-guidelines#loc-laboratory-protocols

We look forward to receiving your revised manuscript.

Kind regards,

Wisit Cheungpasitporn, MD, FACP

University of Mississippi Medical Center

Twitter: @wisit661 Email: wcheungpasitporn@gmail.com 

Academic Editor

PLOS ONE

2. Please provide the full electronic search strategy for at least one database, including any limits used, such that it could be repeated.

3. Please note that according to our submission guidelines (http://journals.plos.org/plosone/s/submission-guidelines), outmoded terms and potentially stigmatizing labels should be changed to more current, acceptable terminology. For example: “Caucasian” should be changed to “white” or “of [Western] European descent” (as appropriate)."

Reviewers' comments:

Reviewer's Responses to Questions

**Comments to the Author**

1. Is the manuscript technically sound, and do the data support the conclusions?

Reviewer #1: Partly

Reviewer #2: Yes

Reviewer #3: Partly

Reviewer #4: Yes

Reviewer #5: Partly

2. Has the statistical analysis been performed appropriately and rigorously? 

Reviewer #1: I Don't Know

Reviewer #2: I Don't Know

Reviewer #3: Yes

Reviewer #4: I Don't Know

Reviewer #5: Yes

3. Have the authors made all data underlying the findings in their manuscript fully available?

Reviewer #1: Yes

Reviewer #2: Yes

Reviewer #3: No

Reviewer #4: Yes

Reviewer #5: No

4. Is the manuscript presented in an intelligible fashion and written in standard English?

Reviewer #1: Yes

Reviewer #2: Yes

Reviewer #3: Yes

Reviewer #4: Yes

Reviewer #5: Yes

5. Review Comments to the Author

Reviewer #1: The authors present the outcomes of a meta-analysis on 5 cohort studies assessing the incidence of Parkinson’s disease in diabetic patients receiving or not receiving thiazolidinedione (TZD) treatment. The outcomes indicate that TZD use leads to a reduced incidence of PD in diabetic patients. However, this conclusion is based on a small number of studies with great (89%) heterogeneity, which seemed to reduce to acceptable levels after removal of 1 study. The topic of this work is timely and of clinical interest. Overall the manuscript is well written, although further methodological detail is required. The review also does not appear to have been prospectively registered. Below I provide my comments per section of the manuscript. I also highlight what information is missing according to the PRISMA 2009 checklist provided by the authors. S=sentence.

Abstract:

-S50: Please specify what is meant by ‘more strict inclusion criteria’.

-PRISMA: Please report: ‘study eligibility criteria’ + ‘limitations’ + ‘registration number’. For synthesising the methods, please also specify what type of studies were included (i.e. retrospective cohort studies) + the outcome of interest (i.e. OR).

Introduction:

-S55 ‘early prominent loss’: I suggest removing the word ‘early’ from this statement, given that much earlier degeneration already occurs at the brainstem level.

-S61-63 ‘Thiazolidinediones…diabets’: Please provide citation.

-S67 ‘Nevertheless, one RCT…’ Please specify this study assessed disease progression in 'early' PD patients.

-S70: Does this statement warrant citations?

-S70-72: Optional -> Consider adding a statement to the introduction about how observational cohort studies can provide useful information besides RCT’s (e.g. in terms of generalizability due to larger and wider-spread samples + perhaps that the outcomes from cohort studies may inform future RCT’s).

Methods:

-S77: Suggest to change ‘were used’ to ‘were screened for eligible studies’

-S79: For replication purposes it would be helpful to provide the exact syntax for the search strategy used per engine in a supplementary document.

-S89: ‘(3) the diagnosis…was definite’ Please specify what is meant with ‘definite’, i.e. according to what diagnostic criteria? Was diagnostic criteria part of the eligibility criteria?

-S92: Please specify what attempts were made to contact corresponding authors to obtain any incomplete data.

-S92: Exclusion criteria (2) ‘studies that did not provide valid data’ can be interpreted in many ways. Please specify or provide examples to indicate what you mean by ‘valid data’.

-S94-97: ‘Data extraction’ should include the incidence of PD to calculate the OR.

-S100: ‘Finally… studies’ This should be placed in the Results section.

-S110-S112: ‘A random-effects…heterogeneity’. This statement is confusing, as to my knowledge either a random-effects or a fixed-effects model can be applied across all studies, not per study as the authors indicate here.

-S122-123: ‘which were not related to the topic’. I suggest to change this to ‘which did not meet eligibility criteria’

-PRISMA: Point 5 is not covered in the manuscript (online protocol/registration).

Results:

-S157-159: ‘With regard…incidence of PD’. I think it is good to highlight again here that only 1 study assessed the use of pioglitazone so that the reader knows the finding should be interpreted with caution.

-The authors could only include a total of 5 studies. I therefore wonder whether this meta-analysis should be termed ‘exploratory’ in the manuscript.

-The authors conduct several sub-group analyses with the aim to determine the source of heterogeneity. To my knowledge, the rationale for performing such sub-analysis is more often to indicate how the main effects change when taking certain factors (e.g. ethnicity, type of drug, etc.) into account, not per se for assessing heterogeneity. Indeed, from these results the authors were not able to make any firm interpretation about heterogeneity. Instead, later when they perform a sensitivity analysis, it soon became clear that the source of heterogeneity is driven by 1 of the 5 studies. Therefore, I wonder whether the sub-group analysis section is currently written with the right scope, and whether the sensitivity analysis should be placed prior to the subgroup analyses section.

-S174-175: ‘Indicating that follow-up contributed little to the heterogeneity’. However, if heterogeneity is 0% for the studies with <10 years follow-up and 92% for studies with >10 years follow-up, does this not indicate that follow up period does have an effect on heterogeneity? The same goes for ethnicity, where the heterogeneity decreased by 50% in the Caucasian cohorts.

-S175-181: The studies are divided based on a quality score on the NOS of <7 or ≥7 points. However, the range across the studies is 6-8 points. I therefore wonder whether this sub-group analysis is informative. In other words, I wonder how much impact a difference in 1-2 points on the NOS has on study quality.

-S202-204: The authors assessed risk of bias using funnel plots. However, in section 10.4.3.1 of the Cochrane handbook (v5.1), it is stated that “As a rule of thumb, test for funnel plot asymmetry should be used only when there are at least 10 studies included in the meta-analysis, because when there are fewer studies the power of the tests is too low to distinguish chance from real asymmetry”. Therefore the inclusion of a funnel plot in this manuscript may lead to incorrect interpretation. This should be clearly mentioned as a limitation in the discussion section and added as a footnote to Figure 3.

-Figure 2: Please add a description in the footnote what the terms ‘Events’, ‘Total’, and ‘M-H’ stand for.

-Figure 3: It now seems that from n=6 records screened, n=44 were excluded.

The horizontal arrows should originate from the previous box. In other words, the n=44 excluded box should originate from the n=50 box, not the n=6 box, etc.

Discussion:

-S229-231: ‘A lower incidence of PD was observed in females than in males’. Could the authors not perform a subgroup analysis to test the impact of sex (male, female) on the OR and heterogeneity?

-S241-242: It seems appropriate to highlight the study by Simuni et al. (2015) again here, which indicated that TZD use did not modify disease progression in early PD

-S247-249: Could the authors report any information about the dosages applied in the 5 included studies, perhaps median (range) dosages? I understand this information may be difficult to retrieve.

Reviewer #2: 1) Need references for the previous studies that have conducted meta-analyzes in the Introduction.

2) Please expand on how the inclusion critieria could be made more strict.

3) Quality assessment of cohort studies: What was the proportion of studies that had disagreement and/or sent out to third reviewer? How were these resolved?

4) How many of the cohorts were studied by the same authors? Any potential lab bias? The average reader may not be familiar with interpreting funnel plots for publication bias. Could the authors please provide explanation in the methods?

5) How many of the cohorts overlapped with previous meta-analyses?

6) The manuscript could benefit with a clearer explanation as to why the meta-analyses conducted in their study is more definitive than the previous studies. Does it all come down to sample size and/or the methodology of the analyses?

Reviewer #3: INTRODUCTION

The study presented by the authors is is one of the first meta-analyses demonstrating the relationship between TZDs use and the incidence of PD in diabetic patients.

There however are several challenges, most prominently the limited number of high quality studies available at this time, and especially the heterogeneity of disease phenotypes and treatment approaches both in DM and PD, turning these kind of analyses into a challenging endeavor at this point in time.

Although the authors do mention the challenges, these are not sufficiently addressed and explained up front.

In addition, one significant clinical data set [Simuni et al. Lancet Neurol. 2015 August ; 14(8): 795–803. doi:10.1016/S1474-4422(15)00144-1], although cited in the introduction (reference 8), is excluded from the meta-analysis for unspecified reasons.

MATERIALS AND METHODS:

Inclusion/Exclusion Criteria:

Given the difficulty to define accurate disease states in DB and PD (definition of ‘definite diagnostics’?), the inclusion criteria by which valid studies were selected regarding disease staging/diagnostics/confounding variables, should have been at least outlined. Ideally, cut-off diagnostic criteria (e.g. Braak staging for PD, diabetes complication severity index for DM) could have been considered for data stratification.

Apart from the importance of disease diagnostic criteria, pharmacological criteria of the different TZDs should be explained and discussed, as the two most used TZDs, pioglitazone and rosiglitazone, differ in both pharmacological as well as medical efficacy, likely influencing both DB and PD disease diagnosis and progression. Similar considerations apply for use of compound medications (sulfonylurea and Metformin) in the studies analyzed.

Data Extraction:

Quality of Meta-analyses heavily relies on the protocols used for data extraction, especially when a number of confounding variables (compounded treatment of different TZDs and medications are at play.

The nature of assessment and data extraction protocols should be explained, including both selection criteria and their rationale.

Handling of conflict, e.g. when disagreement occurred between reviewers regarding Selection and Data Extraction protocols/definition, naming of the third reviewer consulted would be informative.

STATISTICAL ANALYSIS:

It is not clear how and when fixed and random effects models were applied for the analysis.

How was decided which model to use for the individual analyses (Hausman Test?).

One approach to study diversity would be to apply the random-effects model only and then report the expected range of true effects over the populations and interventions sampled. This could be in form of a prediction interval.

The problem with only a few studies to work with is that it is difficult to know how the dispersion actually manifests itself. This section should be expanded.

When based on a small number of studies, estimating the between-studies variance (T2) should be conducted to substantiate the error, as this has important implications for many aspects of the analysis.

Regarding using Odds Ratio (OR) over risk ratio (RR):

It has been shown that RR is more intuitive than the odds ratio (OR) and that OR tend to be interpreted as RR by clinicians, which leads to an overestimate of the effect.

RESULTS:

More detailed legends to the Figures (particularly Figure 2) would be extremely helpful.

Search Results and Study characteristics:

A more detailed flow diagram would help to better understand the selection process and criteria applied.

Meta-analysis of incidence of PD:

The conclusion that follow up duration regarding diagnosis plays no role in study heterogeneity is contradicting PD and DM disease development. These statistical findings give first indications to very broadly defined inclusion / exclusion criteria, especially when considering the observed reduced heterogeneity when excluding the large Lin et al. data set.

Sensitivity analysis and publication bias:

Assessment of the risk of bias (or “quality”) should include the study model constructs being assessed, a definition for each, and reviewer judgment options (high, low, unclear). Finally, the potential influence of this bias and its incorporation into data synthesis, as well as their potential influence on findings of the meta-analysis (“high risk’, ‘low risk’) should be discussed.

DISCUSSION:

Given the confounding variables and limited number of comparable data sets, the unclear stratification of data and the analysis model(s) employed, the meta-analysis as presented does not allow for a conclusion regarding the general benefit of TZDs in PD.

The analytical data presented suggest that that stricter and more precise stratification criteria (most prominently exhibited by the authors post-analysis suggestion of sex-specific differences as contributors to heterogeneity) could allow for inclusion of the Lin et al. study data, and significantly improve the quality of the meta-analysis.

Furthermore, including the Simuni et al. study could increase the power of the meta-analysis significantly.

These improvements and changes would certainly contribute to a more concise and powerful data set, allow better analysis of heterogeneity and it’s sources, and would more accurately describe the general consensus and state of knowledge regarding the use of TZDs as supportive treatment strategy for PD.

Reviewer #4: This paper addresses an interesting but as of yet still uncertain link between use of thiazolidinediones and protective effects on Parkinson's disease. Several preclinical studies have suggested potential benefits of this class of drug and/or its various mechanistic targets including PPARgamma. However, an NIH-funded, Phase II exploratory trial did not find benefit of the drug on disease progression in people with PD. To address this issue, authors have performed a meta-analysis of five identified studies looking at PD risk in diabetics taking thiazolidinedione treatments (glitazones) compared to non-users or users of other diabetic treatments. Authors identify what appears to be a modest protective effect in pooled analyses although heterogeneity across the studies was high. Subgroup analyses did not identify any obvious sources to explain heterogeneity although authors suggest that one of the studies from Lin et al. (which had reported the greatest protective effect) may be, not surprisingly, the primary source of heterogeneity. Removing this study from the analysis reduces the variability and strengthens the statistical significance but also reduces the magnitude of the pooled 'protective' effect. Authors rightfully raise concerns about the limited number of retrospective studies identified as well as other factors (e.g., severity and duration of diabetes across the study populations) that could be a factor. Also, as the diagnosis of motoric PD is likely preceded by an extended period of prodromal disease, issues of reverse causation in studies of this kind (i.e., that diabetics with 'early as yet undiagnosed PD' could be subtly different than diabetics without signs of PD leading to different medication use and overall treatment differences) cannot be ruled out and authors may wish to comment a bit on this aspect. I also am curious about the two studies from Taiwan and whether authors can assess potential for overlapping study population (essentially partially duplicated publication)? Although the two studies in Taiwan reported different outcomes, assessing this potential cohort overlap may be important. Also, in the Newcastle-Ottawa scale assessment, could the authors comment a bit more on those aspects that were not star rated (e.g., Comparability and Adequacy of follow-up)? Do these raise concerns about several of these studies even though overall ratings were considered to be "high quality"? Overall, I think these types of meta-analyses are important for clarifying potential promise of these kinds of treatments although given a number of high-profile recent failures of trials looking at treatments built on reasonably strong epidemiological associations for PD (e.g., blood urate levels, calcium channel blockers), I worry that simple meta-analyses of results will not be sufficient to justify the time and expense of large RCTs and additional biological and/or more detailed retrospective analyses will be needed.

Reviewer #5: 1. This meta-analysis has not been registered online. Please add this point in the limitation.

2. Literature Searches and Search terms are incomplete. This is suboptimal for publication for systematic review. Search terms in Cochrane Central Register of Controlled Trials (CENTRAL), PubMed, Web of Science, and Embase are different. Please attach search terms that were used in each database as supplement for Data source and search strategies in the manuscript. Please provide details search terms in supplementary documents. Please attach syntax used in each database as supplementary.

3. There is substantive heterogeneity in some outcomes. It also is unclear whether the t-statistic is being used for the degrees of freedom in the random effects analysis (i.e., N-1 d.f. not asymptotic [1.96] value multiplied by tau). Please assure that the t-statistic (or Satterthwaite correction) is being used and add that information to the Methods, when the number of studies is small (e.g., < 10). Apply this principle throughout the author's paper. For reference, the authors can refer the article “IntHout J, Ioannidis JP, Borm GF. The Hartung-Knapp-Sidik-Jonkman method for random effects meta-analysis is straightforward and considerably outperforms the standard DerSimonian-Laird method. BMC Medical Research Methodology 2014;14:25.” The issue is the Student t statistic.

4. Authors should discuss the reason of heterogeneity.

5. It will be better to show kappa for the selection and data extraction. Please show the data of kappa of agreement during the systematic searches. How disagreements were solved during the systematic search among two independent reviewers?

6. I recommend the authors apply the ROBINS-I (Risk of Bias in Nonrandomized studies of Interventions) tool in addition to NOS. The authors already applied the Newcastle Ottawa Scale, which is a validated tool and was an acceptable choice. However, to enhance the reproducibility and comparability of this review to future reviews of a similar topic (possibly an update of this review) I recommend including a risk of bias assessment using ROBINS-I, since it is the newest and most robust method of assessing risk of bias in systematic reviews/meta-analyses.

7. Please make the data for this review publicly available, possibly through the Open Science Framework (osf.io). Items to include: list of excluded studies, commands for statistical analysis, spreadsheets or data used for the meta-analyses, etc. Making data publicly available will promote the reproducibility of the review and is best practices for systematic reviews and meta-analyses.

6. PLOS authors have the option to publish the peer review history of their article (what does this mean?). If published, this will include your full peer review and any attached files.

Reviewer #1: No

Reviewer #2: No

Reviewer #3: No

Reviewer #4: No

Reviewer #5: No

---

## [Author Response · Author response to Decision Letter 0]

25 Sep 2019

Dear Editor,

Thank you very much for your letter with regard to our manuscript (PONE-D-19-19116). We sincerely appreciate the editor’s and the reviewers’ comments. We have carefully edited our manuscript according to the reviewers’ comments, which are marked with red font. Our point-by-point responses are summarized below.

In order to enhance the reproducibility of our results, we have deposited our protocol in protocols.io ( the DOI link: http://dx.doi.org/10.17504/protocols.io.7k8hkzw ).

Point 1. 

Respond:

Thank you for your comment. We have revised our manuscript carefully to meet PLOS ONE’s style requirements.

Point 2. 

Please provide the full electronic search strategy for at least one database, including any limits used, such that it could be repeated.

Respond:

Thank you for your comment. We have provided the search strategy in S2 File. 

Point 3. 

Please note that according to our submission guidelines, outmoded terms and potentially stigmatizing labels should be changed to more current, acceptable terminology. For example: “Caucasian” should be changed to “white” or “of [Western] European descent” (as appropriate).

Respond: 

Thank you for your comment. The word “Caucasian” has been changed to “White” in our manuscript. 

Reviewers' comments:

Reviewer #1:

Point 1. 

-S50: Please specify what is meant by ‘more strict inclusion criteria’.

Respond: 

Thank you for your comment. We have specified “more strict inclusion criteria” in our manuscript (Page 2 Line 52,53).

Point 2. 

-PRISMA: Please report: ‘study eligibility criteria’ + ‘limitations’ + ‘registration number’. For synthesising the methods, please also specify what type of studies were included (i.e. retrospective cohort studies) + the outcome of interest (i.e. OR).

Respond: 

Thank you for your comment. We have reported “study eligibility criteria” (Page 2 Line 39-41) and “limitations” (Page 2 Line 49-54). More detailed limitations are available in the Discussion section. We have specified the type of studies which were all retrospective cohort studies (Page 2 Line 44) and the outcome of interest (OR) (Page 2 Line 41,42). Since this meta-analysis was not registered online although we followed a protocol designed for it, we feel sorry that the “registration number” was unavailable (Page 2 Line 49,50). 

Point 3.

-S55‘early prominent loss’: I suggest removing the word ‘early’ from this statement, given that much earlier degeneration already occurs at the brainstem level.

Respond:

Thank you for your comment. We have removed the word “early” (Page 3 Line 57,58).

Point 4.

-S61-63‘Thiazolidinediones…diabets’: Please provide citation.

Respond:

Thank you for your comment. We have provided citation in Page 3 Line 66. 

Point 5.

-S67‘Nevertheless, one RCT…’ Please specify this study assessed disease progression in 'early' PD patients.

Respond:

Thank you for your comment. We have specified that this study assessed patients in early PD (Page 3 Line 71).

Point 6.

-S70: Does this statement warrant citations?

Respond:

Thank you for your comment. We have cited the related references. (Page 3 Line 74).

Point 7.

-S70-72: Optional -> Consider adding a statement to the introduction about how observational cohort studies can provide useful information besides RCT’s (e.g. in terms of generalizability due to larger and wider-spread samples + perhaps that the outcomes from cohort studies may inform future RCT’s).

Respond:

Thank you for your comment. We have added a statement to the “Introduction” about how observational cohort studies can provide useful information besides RCT’s (Page 3 Line 74-77).

Point 8.

-S77: Suggest to change ‘were used’ to ‘were screened for eligible studies’.

Respond:

Thank you for your comment. We have changed “were used” to “were screened for eligible studies”, which should be more appropriate (Page 4 Line 88).

Point 9.

-S79:For replication purposes it would be helpful to provide the exact syntax for the search strategy used per engine in a supplementary document.

Respond:

Thank you for your comment. We have provided the full search strategy used per engine in S2 File.

Point 10.

-S89: ‘(3) the diagnosis…was definite’ Please specify what is meant with ‘definite’, i.e. according to what diagnostic criteria? Was diagnostic criteria part of the eligibility criteria?

Respond:

Thank you for your comment. We must admit that we haven’t expressed our meaning correctly, so we have changed the “(3) the diagnosis…was definite” to “(3) the diagnosis of PD as an outcome” (Page 5 Line 106,107).

Point 11.

-S92: Please specify what attempts were made to contact corresponding authors to obtain any incomplete data.

Respond:

Thank you for your comment. We have specified that we attempted to contact corresponding authors by email to obtain incomplete data in the manuscript (Page 5 Line 110,111).

Point 12.

-S92: Exclusion criteria (2) ‘studies that did not provide valid data’ can be interpreted in many ways. Please specify or provide examples to indicate what you mean by ‘valid data’.

Respond:

Thank you for your comment. We have provided examples for invalid data in Page 5 Line 110. 

Point 13.

-S94-97: ‘Data extraction’ should include the incidence of PD to calculate the OR.

Respond:

Thank you for your comment. We have included the incidence of PD in ‘Data extraction’ to calculate the OR (Page 5 Line 116).

Point 14.

-S100: ‘Finally… studies’ This should be placed in the Results section.

Respond:

Thank you for your comment. We have removed this sentence. And in the Results section, we have the similar description of this result. (Page 7 Line 154,155).

Point 15.

-S110-S112: ‘A random-effects…heterogeneity’. This statement is confusing, as to my knowledge either a random-effects or a fixed-effects model can be applied across all studies, not per study as the authors indicate here.

Respond:

Thank you for your comment. We feel sorry that the statement about random-effects model and random-effects model was confusing. We have revised the manuscript accordingly with more detailed description (Page 6 Line 138-141).

Point 16.

-S122-123: ‘which were not related to the topic’. I suggest to change this to ‘which did not meet eligibility criteria’.

Respond:

Thank you for your comment. We have changed “which were not related to the topic” to “which did not meet eligibility criteria” (Page 6 Line 152, Page 7 Line 153).

Point 17.

-PRISMA: Point 5 is not covered in the manuscript (online protocol/registration).

Respond:

Thank you for your comment. We are sorry that our meta-analysis was not registered online initially. However, we have provided the study protocol which is available in S1 File. And we also deposited our protocol in protocols.io (Page 4 Line 94).

Point 18.

-S157-159: ‘With regard…incidence of PD’. I think it is good to highlight again here that only 1 study assessed the use of pioglitazone so that the reader knows the finding should be interpreted with caution.

Respond:

Thank you for your comment. We have highlighted that only 1 study assessed the use of pioglitazone with regard to different TZDs (Page 11 Line 200,201).

Point 19.

-The authors could only include a total of 5 studies. I therefore wonder whether this meta-analysis should be termed ‘exploratory’ in the manuscript.

Respond:

Thank you for your comment. We have termed this meta-analysis as “an exploratory meta-analysis” in the manuscript.

Point 20.

-The authors conduct several sub-group analyses with the aim to determine the source of heterogeneity. To my knowledge, the rationale for performing such sub-analysis is more often to indicate how the main effects change when taking certain factors (e.g. ethnicity, type of drug, etc.) into account, not per se for assessing heterogeneity. Indeed, from these results the authors were not able to make any firm interpretation about heterogeneity. Instead, later when they perform a sensitivity analysis, it soon became clear that the source of heterogeneity is driven by 1 of the 5 studies. Therefore, I wonder whether the sub-group analysis section is currently written with the right scope, and whether the sensitivity analysis should be placed prior to the subgroup analyses section.

Respond:

Thank you for your comment. We have moved the Sensitivity analysis section prior to the Subgroup analysis section. And we also revised the interpretation of the Subgroup analysis according to your suggestion.

Point 21.

-S174-175: ‘Indicating that follow-up contributed little to the heterogeneity’. However, if heterogeneity is 0% for the studies with <10 years follow-up and 92% for studies with >10 years follow-up, does this not indicate that follow up period does have an effect on heterogeneity? The same goes for ethnicity, where the heterogeneity decreased by 50% in the Caucasian cohorts.

Respond:

Thank you for your comment. We agree that follow-up duration and ethnicity should have an impact on heterogeneity. We have revised the manuscript accordingly (Page 11 Line 207-209, Line213-216). 

Point 22.

-S175-181: The studies are divided based on a quality score on the NOS of <7 or ≥7 points. However, the range across the studies is 6-8 points. I therefore wonder whether this sub-group analysis is informative. In other words, I wonder how much impact a difference in 1-2 points on the NOS has on study quality.

Respond:

Thank you for your comment. According to Reviewer #5’s suggestion, we have applied the ROBINS-I to assess the quality of studies (Page 5 Line 120-128, Page 6 Line 129-130). And the subgroup analysis based on NOS has been removed.

Point 23.

-S202-204: The authors assessed risk of bias using funnel plots. However, in section 10.4.3.1 of the Cochrane handbook (v5.1), it is stated that “As a rule of thumb, test for funnel plot asymmetry should be used only when there are at least 10 studies included in the meta-analysis, because when there are fewer studies the power of the tests is too low to distinguish chance from real asymmetry”. Therefore the inclusion of a funnel plot in this manuscript may lead to incorrect interpretation. This should be clearly mentioned as a limitation in the discussion section and added as a footnote to Figure 3.

Respond:

Thank you for your comment. We have mentioned it as a limitation in the Discussion section (Page14 Line 265-267, Page15 Line 268) and added a footnote to Figure 3 (Page 13 Line 225-227).

Point 24.

-Figure 2: Please add a description in the footnote what the terms ‘Events’, ‘Total’, and ‘M-H’ stand for.

Respond:

Thank you for your comment. We have added a description in the footnote about the meanings of “Events”, “Total”, and “M-H” (Page 10 Line184-186).

Point 25.

-Figure 3: It now seems that from n=6 records screened, n=44 were excluded.

The horizontal arrows should originate from the previous box. In other words, the n=44 excluded box should originate from the n=50 box, not the n=6 box, etc.

Respond:

Thank you for your comment. We have revised our flow diagram.

Point 26.

-S229-231: ‘A lower incidence of PD was observed in females than in males’. Could the authors not perform a subgroup analysis to test the impact of sex (male, female) on the OR and heterogeneity?

Respond:

Thank you for your comment. Performing a subgroup analysis based on sex (male, female) is meaningful. However, the data was not provided, which makes it impossible to perform a subgroup analysis to test the impact of sex on the OR and heterogeneity.

Point 27.

-S241-242: It seems appropriate to highlight the study by Simuni et al. (2015) again here, which indicated that TZD use did not modify disease progression in early PD.

Respond:

Thank you for your comment. We have highlighted the study by Simuni et al. (2015) again in the Discussion section (Page 14 Line 259,260).

Point 28.

-S247-249: Could the authors report any information about the dosages applied in the 5 included studies, perhaps median (range) dosages? I understand this information may be difficult to retrieve.

Respond:

Thank you for your comment. We agree that reporting information about the dosages would improve our manuscript. However, it is a pity that detailed dosages have not been found. 

Reviewer #2:

Point 1.

1) Need references for the previous studies that have conducted meta-analyzes in the Introduction.

Respond:

Thank you for your comment. We did not find any related meta-analysis regarding this topic. Therefore, we believe our meta-analysis would provide valuable information concerning the association between TZDs use and incidence of PD (Page 4 Line 81-83).

Point 2.

2) Please expand on how the inclusion critieria could be made more strict.

Respond:

Thank you for your comment. We have specified “more strict inclusion criteria” in our manuscript (Page 2 Line 52,53).

Point 3.

3) Quality assessment of cohort studies: What was the proportion of studies that had disagreement and/or sent out to third reviewer? How were these resolved?

Respond:

Thank you for your comments. We have disagreement on one of the five studies. As we mentioned in Page 6 Line 131, disagreements were solved through discussion with a third reviewer (Jiali Pu). 

Point 4.

4) How many of the cohorts were studied by the same authors? Any potential lab bias? The average reader may not be familiar with interpreting funnel plots for publication bias. Could the authors please provide explanation in the methods?

Respond:

Thank you for your comments. Two investigators (Yueli Zhu, Yanxing Chen) independently evaluated all included studies and performed all statistical analysis. And there should be no potential lab bias. We mentioned in the manuscript about the interpretation of funnel plots for publication bias (Page 6 Line 146,147).

Point 5.

5) How many of the cohorts overlapped with previous meta-analyses?

Respond:

Thank you for your comment. As far as we know, no meta-analysis has been conducted to explore the association between TZDs use and the incidence of PD in diabetic patients so far.

Point 6.

6) The manuscript could benefit with a clearer explanation as to why the meta-analyses conducted in their study is more definitive than the previous studies. Does it all come down to sample size and/or the methodology of the analyses?

Respond:

Thank you for your comment. According to the Cochrane Handbook for Systematic Reviews of Interventions, meta-analysis is used to summarize the results of independent studies and answer a specific research question. By combining information from all relevant studies, meta-analyses can provide more precise estimates of the effects of health care than those derived from the individual studies included within a review. They also facilitate investigations of the consistency of evidence across studies, and the exploration of differences across studies. As far as we know, this is the first meta-analysis conducted to assess the association between TZDs use and the incidence of PD in diabetic patients (Page 4 Line 82,83).

Reviewer #3:

Point 1.

There however are several challenges, most prominently the limited number of high quality studies available at this time, and especially the heterogeneity of disease phenotypes and treatment approaches both in DM and PD, turning these kind of analyses into a challenging endeavor at this point in time. Although the authors do mention the challenges, these are not sufficiently addressed and explained up front.

Respond:

Thank you for your comment. We have explained these challenges in the Introduction section (Page 3 Line 79, Page 4 Line 80,81).

Point 2.

In addition, one significant clinical data set [Simuni et al. Lancet Neurol. 2015 August ; 14(8): 795–803. doi:10.1016/S1474-4422(15)00144-1], although cited in the introduction (reference 8), is excluded from the meta-analysis for unspecified reasons.

Respond:

Thank you for your comment. The purpose of the current meta-analysis is to explore the association between TZDs use and the incidence of PD in diabetic patients. However, the study conducted by Simuni et al. was designed to assess whether TZDs use could slow the disease progression in early PD patients without diabetes. So this study was excluded from this meta-analysis. And we have mentioned it in the flow diagram (Fig 1).

Point 3.

Given the difficulty to define accurate disease states in DB and PD (definition of ‘definite diagnostics’?), the inclusion criteria by which valid studies were selected regarding disease staging/diagnostics/confounding variables, should have been at least outlined. Ideally, cut-off diagnostic criteria (e.g. Braak staging for PD, diabetes complication severity index for DM) could have been considered for data stratification.

Respond:

Thank you for your comment. We have changed the “(3) the diagnosis…was definite” to “(3) the diagnosis of PD as an outcome” (Page 5 Line 106,107). It is true that ideally Braak staging for PD and diabetes complication severity index for DM could be used for data stratification, which would provide additional information for this meta-analysis. However, information regarding disease staging was not available. Therefore, we only analyzed the incidence of PD.

Point 4.

Apart from the importance of disease diagnostic criteria, pharmacological criteria of the different TZDs should be explained and discussed, as the two most used TZDs, pioglitazone and rosiglitazone, differ in both pharmacological as well as medical efficacy, likely influencing both DB and PD disease diagnosis and progression. Similar considerations apply for use of compound medications (sulfonylurea and Metformin) in the studies analyzed.

Respond:

Thanks for your valuable suggestion. We also believe that the use of different TZDs should be discussed, which is important to improve the quality of our manuscript. Regrettably, it was difficult to analyze since some studies did not specify TZDs ( pioglitazone or rosiglitazone). It was also hard to consider the use of compound medications (sulfonylurea and Metformin) due to incomplete data.

Point 5.

Quality of Meta-analyses heavily relies on the protocols used for data extraction, especially when a number of confounding variables (compounded treatment of different TZDs and medications are at play. The nature of assessment and data extraction protocols should be explained, including both selection criteria and their rationale.

Respond:

Thank you for your comment. We have provided the study protocol and it is available in S1 File.

Point 6.

Handling of conflict, e.g. when disagreement occurred between reviewers regarding Selection and Data Extraction protocols/definition, naming of the third reviewer consulted would be informative.

Respond:

Thank you for your comment. Disagreements were solved through discussion with a third reviewer if necessary (Jiali Pu). We have mentioned it in our manuscript (Page 5 Line 116,117).

Point 7.

It is not clear how and when fixed and random effects models were applied for the analysis. How was decided which model to use for the individual analyses (Hausman Test?). One approach to study diversity would be to apply the random-effects model only and then report the expected range of true effects over the populations and interventions sampled. This could be in form of a prediction interval. The problem with only a few studies to work with is that it is difficult to know how the dispersion actually manifests itself. This section should be expanded.

Respond:

Thank you for your comments. Since the Cochrane Collaboration’s Review Manager 5.3 software was used to analyze data, we have performed both the chi-square test (assessing the p-value) and the I 2 statistic to estimate the heterogeneity. P < 0.1 and I2 > 50% for heterogeneity were considered significantly different, then a random-effects model was used to calculate the pooled ORs. And outcome data were pooled using a fixed-effects model with insignificant heterogeneity. And the between-study variance was estimated using tau-squared (t2 or Tau2 ) statistic in the random-effects model. Besides, we are unable to use a prediction interval because we conducted the meta-analyses with Review Manager. In order to find out the potential sources of heterogeneity, sensitivity analysis and subgroup analysis were all performed. We have revised and expanded the Statistical analysis section.

Point 8.

When based on a small number of studies, estimating the between-studies variance (T2) should be conducted to substantiate the error, as this has important implications for many aspects of the analysis.

Respond:

Thank you for your comment. We have added the use of tau-squared (t2 or Tau2 ) statistic to estimate the between-study variance in our manuscript (Page 6 Line 142 and Page 10 Line 174).

Point 9.

It has been shown that RR is more intuitive than the odds ratio (OR) and that OR tend to be interpreted as RR by clinicians, which leads to an overestimate of the effect.

Respond:

Thank you for your comment. On the basis of the Cochrane handbook, if the odds ratio (OR) is misinterpreted as a risk ratio (RR), problems may arise. The OR will be larger than the RR for interventions that increase the chances of events, so the misinterpretation will tend to overestimate the intervention effect. For interventions that reduce the chances of events, the OR will be smaller than the RR, so the misinterpretation overestimates the effect of the intervention as well. The non equivalence of the RR and OR does not indicate that either is wrong and both are entirely valid ways of describing an intervention effect. In the current study, we used odds ratios (OR) with 95% CI.

Point 10.

More detailed legends to the Figures (particularly Figure 2) would be extremely helpful.

Respond:

Thank you for your comment. We have provided more detailed legends to the Fig 2 and Fig 3 (Page 10 Line180-186, Page 13 Line 225-227).

Point 11.

A more detailed flow diagram would help to better understand the selection process and criteria applied.

Respond:

Thank you for your comment. We have provided a more detailed flow diagram (Fig1).

Point 12.

The conclusion that follow up duration regarding diagnosis plays no role in study heterogeneity is contradicting PD and DM disease development. These statistical findings give first indications to very broadly defined inclusion / exclusion criteria, especially when considering the observed reduced heterogeneity when excluding the large Lin et al. data set.

Respond:

Thank you for your comment. We feel sorry for the improper wording. Regarding follow-up duration, I2 for studies with follow-up duration < 10 years was 0, and it increased to 92% in studies with follow-up duration ≥ 10 years, indicating that follow-up duration might be one of the sources of heterogeneity. Therefore, follow-up duration should have an effect on heterogeneity. And we have revised our manuscript in Page 11 Line 213-216.

Point 13.

Assessment of the risk of bias (or “quality”) should include the study model constructs being assessed, a definition for each, and reviewer judgment options (high, low, unclear). Finally, the potential influence of this bias and its incorporation into data synthesis, as well as their potential influence on findings of the meta-analysis (“high risk’, ‘low risk’) should be discussed.

Respond:

Thank you for your comment. According to the Reviewer #5’s suggestion, we have applied the ROBINS-I to assess the quality of studies. And it has been detailed in our manuscript (Page 5 Line120-128, Page 6 Line 129,130).

Point 14.

Given the confounding variables and limited number of comparable data sets, the unclear stratification of data and the analysis model(s) employed, the meta-analysis as presented does not allow for a conclusion regarding the general benefit of TZDs in PD.

Respond:

Thanks for your nice suggestion. Given the confounding variables and only 5 included studies, we have changed the title of this meta-analysis to “an exploratory meta-analysis”. And we have listed some limitations of our study in the Discussion section (Page 15 Line 268-275).

Point 15.

The analytical data presented suggest that stricter and more precise stratification criteria (most prominently exhibited by the authors post-analysis suggestion of sex-specific differences as contributors to heterogeneity) could allow for inclusion of the Lin et al. study data, and significantly improve the quality of the meta-analysis.

Respond:

Thank you for your comment. We initially had considered to include sex-specific differences for data stratification. However, the gender differences with regard to the incidence of PD was not available from the study conducted by Lin et al.

Point 16. 

Furthermore, including the Simuni et al. study could increase the power of the meta-analysis significantly.

Respond:

Thank you for your comment. As we know, the study conducted by Simuni et al. is of high quality and the purpose is to explore whether TZD use could modify disease progression in early PD, not the incidence. But our meta-analysis was conducted to evaluate the efficacy of TZDs in reducing PD risk among diabetic patients. Hence, we excluded the study conducted by Simuni et al.

Reviewer #4:

Point 1.

Also, as the diagnosis of motoric PD is likely preceded by an extended period of prodromal disease, issues of reverse causation in studies of this kind (i.e., that diabetics with 'early as yet undiagnosed PD' could be subtly different than diabetics without signs of PD leading to different medication use and overall treatment differences) cannot be ruled out and authors may wish to comment a bit on this aspect. 

Respond:

Thanks for your comment. We have mentioned it as a limitation in the Discussion section (Page15 Line 275-279).

Point 2.

 I also am curious about the two studies from Taiwan and whether authors can assess potential for overlapping study population (essentially partially duplicated publication)? Although the two studies in Taiwan reported different outcomes, assessing this potential cohort overlap may be important.

Respond:

Thanks for your comment. According to the Cochrane handbook, the production of multiple publications from single studies can lead to bias in a number of ways. It is not always obvious that multiple publications come from a single study, and one set of study participants may be included in an analysis twice. The inclusion of duplicated data may therefore lead to overestimation of intervention effects.

It is true that the two studies from Taiwan may involve duplicated patients. But they have reported different outcomes. Besides, the shape of funnel plots conducted to assess the latent publication bias was symmetric, although it was a limitation that our included studies were less than ten.

Point 3.

Also, in the Newcastle-Ottawa scale assessment, could the authors comment a bit more on those aspects that were not star rated (e.g., Comparability and Adequacy of follow-up)? Do these raise concerns about several of these studies even though overall ratings were considered to be "high quality"? 

Respond:

Thanks for your comment. According to the Reviewer #5’s suggestion, we have applied the ROBINS-I to assess the quality of studies. And it has been detailed in our manuscript (Page 5 Line120-128, Page 6 Line 129-130).

Point 4.

Overall, I think these types of meta-analyses are important for clarifying potential promise of these kinds of treatments although given a number of high-profile recent failures of trials looking at treatments built on reasonably strong epidemiological associations for PD (e.g., blood urate levels, calcium channel blockers), I worry that simple meta-analyses of results will not be sufficient to justify the time and expense of large RCTs and additional biological and/or more detailed retrospective analyses will be needed.

Respond:

Thanks for your comment and we agree with your suggestion. Further biological studies are needed to specify the potential neuroprotective mechanism of TZDs in PD. And future prospective studies with larger cohorts are of great value to confirm these results, which would guide the designation of RCTs. We have revised our manuscript accordingly (Page 15 Line 269-275).

Reviewer #5:

Point 1.

1. This meta-analysis has not been registered online. Please add this point in the limitation.

Respond:

Thanks for your comment. We feel sorry that we did not register online but we have followed a protocol designed for the meta-analysis which is available in S1 File. And we have added this point in the limitation (Page 14 Line 264,265).

Point 2.

2. Literature Searches and Search terms are incomplete. This is suboptimal for publication for systematic review. Search terms in Cochrane Central Register of Controlled Trials (CENTRAL), PubMed, Web of Science, and Embase are different. Please attach search terms that were used in each database as supplement for Data source and search strategies in the manuscript. Please provide details search terms in supplementary documents. Please attach syntax used in each database as supplementary.

Respond:

Thanks for your comment. We are very sorry for our negligence of providing search strategies. Now we have provided detailed search strategies in S2 File.

Point 3.

3. There is substantive heterogeneity in some outcomes. It also is unclear whether the t-statistic is being used for the degrees of freedom in the random effects analysis (i.e., N-1 d.f. not asymptotic [1.96] value multiplied by tau). Please assure that the t-statistic (or Satterthwaite correction) is being used and add that information to the Methods, when the number of studies is small (e.g., < 10). Apply this principle throughout the author's paper. For reference, the authors can refer the article “IntHout J, Ioannidis JP, Borm GF. The Hartung-Knapp-Sidik-Jonkman method for random effects meta-analysis is straightforward and considerably outperforms the standard DerSimonian-Laird method. BMC Medical Research Methodology 2014;14:25.” The issue is the Student t statistic.

Respond:

Thanks for your nice suggestion. It has been raised that DerSimonian-Laird (DL) method is not suitable when the number of studies in the meta-analysis is small [1]. Hence, in the current study, we used Mantel-Haenszel method. We also tried to use the recommended HKSJ method for random effects in our meta-analysis. The pooled OR was 0.7 (HKSJ 95% CI, 0.45-1.09; t = -2.23; P-value = 0.09; df = 4), indicating that the TZDs use was not associated with reduced risk of PD. As we know, however, HKSJ analysis method has some limitations [1]. When one of the studies in the meta-analysis was substantially larger than the other ones, the error rates will be maximal [1]. In our meta-analysis, the study with the largest sample was 10-times larger than the smallest one, which would underestimate the protective effects of TZDs by using the HKSJ analysis method. According to the Cochrane handbook, Mantel-Haenszel methods have been shown to have better statistical properties when there are few events. Hence, we have used the Mantel-Haenszel method. Besides, after removing the study by Lin et al., we used the fixed-effects model and a significant association of TZDs use with lower incidence of PD in diabetic patients was observed (OR, 0.84; 95% CI, 0.74 to 0.94; p = 0.003; I2 = 10%; Fig 2B). In conclusion ,we hold the opinion that the TZDs use should be related to the reduced risk of PD and further studies are warranted.

[1]. IntHout J, Ioannidis JP, Borm GF. The Hartung-Knapp-Sidik-Jonkman method for random effects meta-analysis is straightforward and considerably outperforms the standard DerSimonian-Laird method. BMC Med Res Methodol. 2014;14:25.

Point 4.

4. Authors should discuss the reason of heterogeneity.

Respond:

Thanks for your comment. We have discussed the reason of heterogeneity in our manuscript (Page 13 Line 240-242, Page 14 Line 243-252).

Point 5.

5. It will be better to show kappa for the selection and data extraction. Please show the data of kappa of agreement during the systematic searches. How disagreements were solved during the systematic search among two independent reviewers?

Respond:

Thanks for your comments. We have provided the kappa value during the systematic searches in our manuscript (Page7 Line155,156). Disagreements were solved through discussion with a third reviewer if necessary (Jiali Pu).

Point 6.

6. I recommend the authors apply the ROBINS-I (Risk of Bias in Nonrandomized studies of Interventions) tool in addition to NOS. The authors already applied the Newcastle Ottawa Scale, which is a validated tool and was an acceptable choice. However, to enhance the reproducibility and comparability of this review to future reviews of a similar topic (possibly an update of this review) I recommend including a risk of bias assessment using ROBINS-I, since it is the newest and most robust method of assessing risk of bias in systematic reviews/meta-analyses.

Respond:

Thanks for your nice suggestion. And we have assessed risk of bias using ROBINS-I.

Point 7.

7. Please make the data for this review publicly available, possibly through the Open Science Framework (osf.io). Items to include: list of excluded studies, commands for statistical analysis, spreadsheets or data used for the meta-analyses, etc. Making data publicly available will promote the reproducibility of the review and is best practices for systematic reviews and meta-analyses.

Respond:

Thanks for your comment. We agree that it is necessary to make our data publicly available to enhance the reproducibility our results. According to the editor’s comment, We have deposited our protocol in protocols.io ( DOI link: http://dx.doi.org/10.17504/protocols.io.7k8hkzw ). Besides, we have provided the study protocol which is available in S1 File. Readers could obtain information about our data online.

Once again, we greatly appreciate both your help and that of the reviewers concerning improvements to this paper. We believe the manuscript has been improved satisfactorily and hope it will be accepted for publication in PLOS ONE.

Sincerely,

Baorong Zhang, M.D.

---

## [Decision Letter · Decision Letter 1]

9 Oct 2019

Decreased risk of Parkinson’s disease in diabetic patients with thiazolidinediones therapy: An exploratory meta-analysis

PONE-D-19-19116R1

Dear Dr. Baorong Zhang,

We are pleased to inform you that your manuscript has been judged scientifically suitable for publication and will be formally accepted for publication once it complies with all outstanding technical requirements.

With kind regards,

Wisit Cheungpasitporn, MD, FACP, FASN

University of Mississippi Medical Center

Twitter: @wisit661 Email: wcheungpasitporn@gmail.com 

Academic Editor

PLOS ONE

Additional Editor Comments:

I want to commend the authors on their superb efforts to revise the manuscript according to all reviewers’ suggestions. The quality of the manuscript has improved substantially. 

Reviewers' comments:

Reviewer's Responses to Questions

**Comments to the Author**

1. If the authors have adequately addressed your comments raised in a previous round of review and you feel that this manuscript is now acceptable for publication, you may indicate that here to bypass the “Comments to the Author” section, enter your conflict of interest statement in the “Confidential to Editor” section, and submit your "Accept" recommendation.

Reviewer #2: All comments have been addressed

Reviewer #3: All comments have been addressed

Reviewer #5: All comments have been addressed

2. Is the manuscript technically sound, and do the data support the conclusions?

Reviewer #2: Yes

Reviewer #3: Yes

Reviewer #5: Yes

3. Has the statistical analysis been performed appropriately and rigorously? 

Reviewer #2: Yes

Reviewer #3: Yes

Reviewer #5: Yes

4. Have the authors made all data underlying the findings in their manuscript fully available?

Reviewer #2: Yes

Reviewer #3: Yes

Reviewer #5: Yes

5. Is the manuscript presented in an intelligible fashion and written in standard English?

Reviewer #2: Yes

Reviewer #3: Yes

Reviewer #5: Yes

6. Review Comments to the Author

Reviewer #2: (No Response)

Reviewer #3: I thank the author(s) for addressing my questions and concerns.

Given the changes made and the major concerns being addressed, I have no further reservations regarding publication of the revised version of the manuscript.

Reviewer #5: All my concerns have been fully elucidated, missing sections and analyses have been completed. Finally, comprehension errors have been corrected.

7. PLOS authors have the option to publish the peer review history of their article (what does this mean?). If published, this will include your full peer review and any attached files.

Reviewer #2: No

Reviewer #3: No

Reviewer #5: No

---

## [Editor Report · Acceptance letter]

14 Oct 2019

PONE-D-19-19116R1 

Decreased risk of Parkinson’s disease in diabetic patients with thiazolidinediones therapy: An exploratory meta-analysis 

Dear Dr. Zhang:

I am pleased to inform you that your manuscript has been deemed suitable for publication in PLOS ONE. Congratulations! Your manuscript is now with our production department. 

With kind regards,

on behalf of

Dr. Wisit Cheungpasitporn 

Academic Editor

PLOS ONE